# Type 2 diabetes mellitus accelerates brain aging and cognitive decline: Complementary findings from UK Biobank and meta-analyses

Botond Antal[1,2†], Liam P McMahon[1,2], Syed Fahad Sultan[3], Andrew Lithen[1,2], Deborah J Wexler[4], Bradford Dickerson[2,5], Eva-Maria Ratai[2], Lilianne R Mujica-Parodi[1,2,6]*

[1]Department of Biomedical Engineering, Stony Brook University, Stony Brook, United States; [2]Athinoula A. Martinos Center for Biomedical Imaging, Massachusetts General Hospital and Harvard Medical School, Charlestown, United States; [3]Department of Computer Science, Stony Brook University, Stony Brook, United States; [4]Diabetes Center, Massachusetts General Hospital and Harvard Medical School, Boston, United States; [5]Department of Neurology, Massachusetts General Hospital and Harvard Medical School, Boston, United States; [6]Department of Neurology, Stony Brook University School of Medicine, Stony Brook, United States

*For correspondence:
mujica@lcneuro.org

†is the sole first author

## Abstract

**Background:** Type 2 diabetes mellitus (T2DM) is known to be associated with neurobiological and cognitive deficits; however, their extent, overlap with aging effects, and the effectiveness of existing treatments in the context of the brain are currently unknown.

**Methods:** We characterized neurocognitive effects independently associated with T2DM and age in a large cohort of human subjects from the UK Biobank with cross-sectional neuroimaging and cognitive data. We then proceeded to evaluate the extent of overlap between the effects related to T2DM and age by applying correlation measures to the separately characterized neurocognitive changes. Our findings were complemented by meta-analyses of published reports with cognitive or neuroimaging measures for T2DM and healthy controls (HCs). We also evaluated in a cohort of T2DM-diagnosed individuals using UK Biobank how disease chronicity and metformin treatment interact with the identified neurocognitive effects.

**Results:** The UK Biobank dataset included cognitive and neuroimaging data (N = 20,314), including 1012 T2DM and 19,302 HCs, aged between 50 and 80 years. Duration of T2DM ranged from 0 to 31 years (mean 8.5 ± 6.1 years); 498 were treated with metformin alone, while 352 were unmedicated. Our meta-analysis evaluated 34 cognitive studies (N = 22,231) and 60 neuroimaging studies: 30 of T2DM (N = 866) and 30 of aging (N = 1088). Compared to age, sex, education, and hypertension-matched HC, T2DM was associated with marked cognitive deficits, particularly in *executive functioning* and *processing speed*. Likewise, we found that the diagnosis of T2DM was significantly associated with gray matter atrophy, primarily within the *ventral striatum*, *cerebellum*, and *putamen*, with reorganization of brain activity (decreased in the *caudate* and *premotor cortex* and increased in the *subgenual area*, *orbitofrontal cortex, brainstem,* and *posterior cingulate cortex*). The structural and functional changes associated with T2DM show marked overlap with the effects correlating with age but appear earlier, with disease duration linked to more severe neurodegeneration. Metformin treatment status was not associated with improved neurocognitive outcomes.

**Conclusions:** The neurocognitive impact of T2DM suggests marked acceleration of normal brain aging. T2DM gray matter atrophy occurred approximately 26% ± 14% faster than seen with normal aging; disease duration was associated with increased neurodegeneration. Mechanistically, our results suggest a neurometabolic component to brain aging. Clinically, neuroimaging-based biomarkers may provide a valuable adjunctive measure of T2DM progression and treatment efficacy based on neurological effects.

**Funding:** The research described in this article was funded by the W. M. Keck Foundation (to LRMP), the White House Brain Research Through Advancing Innovative Technologies (BRAIN) Initiative (NSFNCS-FR 1926781 to LRMP), and the Baszucki Brain Research Fund (to LRMP). None of the funding sources played any role in the design of the experiments, data collection, analysis, interpretation of the results, the decision to publish, or any aspect relevant to the study. DJW reports serving on data monitoring committees for Novo Nordisk. None of the authors received funding or in-kind support from pharmaceutical and/or other companies to write this article.

## Editor's evaluation

This work emphasizes the role of diabetes in brain aging and cognitive functions that is a critical gap that needs to be filled due to the increasing trend in the prevalence of diabetes around the world. This article provides valuable information about specific brain regions altered during aging and diabetes. Further, this article reports how T2DM accelerates the aging-associated decline in cognition and brain function. Extensive analysis of human datasets and comparison with published data from other researchers support the conclusion of this study; however, certain diabetic interventions that do not rescue brain damage need further validation.

## Introduction

Approximately 6.6% of the global population carries a diagnosis of type 2 diabetes mellitus (T2DM) (*Kaiser et al., 2018*). Patients with T2DM are at greater risk for developing dementia and Alzheimer's disease (AD) and have been reported to exhibit inferior cognitive performance when compared to age-matched healthy controls (HCs) (*Moheet et al., 2015*). Several human neuroimaging studies have linked T2DM with brain atrophy and cognition (*Moheet et al., 2015*; *Callisaya et al., 2019*; *Gold et al., 2007*; *Mankovsky et al., 2018*; *Manschot et al., 2006*); recent research suggested that T2DM resulted in a more rapid rate of cognitive decline than typically associated with natural aging (*Rawlings et al., 2014*; *Zilliox et al., 2016*; *Pelimanni and Jehkonen, 2018*).

Despite strong preliminary evidence linking T2DM to neurological and cognitive decline, few patients with T2DM undergo a comprehensive neurocognitive evaluation as part of their clinical care (*Zilliox et al., 2016*; *Biessels et al., 2006*; *Ninomiya, 2014*). This may reflect the fact that T2DM diagnosis often occurs in middle age, hindering dissociation of patients' cognitive changes from normal aging. Several studies published to date focused on the neurocognitive effects of T2DM including age-matched participants. However, because none has compared lifespan neurological changes to those experienced by equivalently aged patients with T2DM, it is currently unknown whether neurocognitive effects represent a T2DM-specific neurodegenerative pathway or the exacerbation of typical brain aging. Moreover, there remain limited data (*Tuligenga, 2015*) evaluating the impact of chronicity or role of effective treatment in the progression of cognitive and neurological decline.

Routine clinical protocols typically focus on peripheral biomarkers (e.g., blood glucose and insulin levels, body fat percentage) as diagnostic modalities for T2DM. However, the neurological effects of T2DM may reveal themselves many years before they can be detected by peripheral markers (*Callisaya, 2019*; *Zilliox et al., 2016*). As such, by the time T2DM is diagnosed and treated by standard measures, patients may have already sustained irreversible brain damage. Thus, there are direct clinical implications with respect to defining the neurocognitive impact of T2DM and to determine how these negative sequelae might be prevented or treated (*Kaiser et al., 2018*).

Given these unknowns and their clinical importance, here we focus on addressing three questions. First, we establish T2DM neurocognitive effects compared to age, sex, education, and hypertension-matched HCs. To do so, we leverage the robust statistical power made possible by UK Biobank (*Sudlow et al., 2015*), the largest (N = 20,314, ages 50–80 years) neurocognitive lifespan dataset to

date. The UK Biobank results are then compared to a meta-analysis of the published literature (34 cognitive studies, 60 neuroimaging studies) to assess convergence. Second, we ask whether changes in the brain observed in T2DM represent accelerated aging or a nonaging-related degenerative pathway specific to T2DM. Third, we test whether T2DM chronicity exacerbates, and medication status ameliorates, the progression of neurocognitive effects.

## Methods
### Analysis of UK Biobank dataset (N = 20,314)
#### General overview
UK Biobank data were analyzed for both cognitive and neuroimaging data. Datafield identifiers for all utilized features are shown in *Supplementary file 1*. The primary factor of interest was T2DM, which we dissociated from age-related effects by age matching T2DM and HC. To permit comparison of T2DM-specific effects to age-specific effects, we also assessed the same neurocognitive variables with age as a factor of interest from samples that excluded patients diagnosed with T2DM. To control for potential neurocognitive confounds, T2DM and HC were exact pairwise matched for not only age, but also sex, education, and hypertension status. T2DM status was assessed based on self-reported diagnosis by doctor. We considered education as a binary variable based on possession of a college degree. Hypertension was quantified using measured blood pressure values: all individuals with systolic blood pressure >140 mmHg or diastolic blood pressure >90 mmHg were labeled with hypertension (*Chobanian et al., 2003*). To exclude potential confounding effects due to menopausal transition, particularly relevant to the youngest age group in our sample (age cutoff >50 years), we excluded all female subjects who did not report menopause or if they reported ongoing hormone therapy (*Maki and Henderson, 2016*). To minimize the number of individuals with type 1 diabetes in our sample, rather than type 2 diabetes, we only included diabetic individuals with a self-reported age of onset ≥ 40 years (*Thomas et al., 2018*).

We fitted linear regression models to neurocognitive variables and quantified associated effects as the maximum likelihood point estimates and confidence intervals (95% CIs) of the corresponding regression coefficients. T2DM was accounted for as a binary factor with two states corresponding to HCs and individuals diagnosed with T2DM, whereas age was considered as a linear continuous factor with increments in years. The latter was justified given the linear trends we observed across age in all modalities (*Appendix 1—figures 4–6*). Regression models were fit using the Statsmodels Python library (*Seabold and Perktold, 2010*). To account for multiple comparisons, we applied Bonferroni correction to cognitive and structural results, and adjusted for false discovery rate (FDR) in our brain activation results; we report adjusted p-values exclusively.

To determine whether T2DM neurocognitive effects suggested an acceleration of typical aging trajectories, versus nonaging-related degenerative pathways specific to T2DM, we compared the progression of neurodegeneration seen in T2DM to that seen in relation to age across brain regions and cognitive domains using bivariate Pearson correlations.

#### Cognition
Data on five cognitive domains for 18,829 participants (T2DM: N = 914, HC: N = 17,915) were extracted from the UK Biobank dataset, including *abstract reasoning*, *executive function*, *processing speed*, *reaction time*, and *numeric memory* (~2–3 s). Exact sample sizes varied across cognitive domains based on data availability, and therefore are noted separately for each result (*Appendix 1—figures 1C and 2C*). We employed linear regression and considered the maximum likelihood estimates of coefficients belonging to age and T2DM to estimate their associations with performance in each of the five domains. Effect sizes in cognition were quantified as percentages by dividing the estimated β coefficient and 95% CIs of the factor of interest with the average performance of HC.

#### Brain structure
Using structural MRI data from the UK Biobank dataset, we assessed the effects associated with T2DM (T2DM: N = 821, HC: N = 821, *Appendix 1—figure 1A*) compared to non-T2DM-specific age-related effects (N = 4775, *Appendix 1—figure 2A*) on atrophy of gray matter volume , calculated as mm³ (*Callisaya et al., 2019*) for the whole brain as well as for 139 anatomical regions. For region-specific

analyses, we coarse-grained the default unilateral parcellation of 139 regions provided by UK Biobank into 45 bilateral regions and normalized gray matter volumes for head size. We applied linear regression and quantified atrophy in each anatomical region as a relative percentage change in average gray matter volume by dividing the estimated regression coefficients and 95% CIs of the factor of interest by the average gray matter volume of HC.

### Brain function

Using functional MRI data from the UK Biobank dataset, we assessed the effects occurring with T2DM (T2DM: N = 646; HC: N = 646, *Appendix 1—figure 1B*) compared to non-T2DM-specific age-related effects (N = 2250, *Appendix 1—figure 2B*) on resting-state brain activity. Data were accessed already preprocessed by UK Biobank according to their standard pipelines (*Alfaro-Almagro et al., 2018*). After transforming functional images to Montreal Neurological Institute (MNI) space, we performed spatial smoothing with a full width at half maximum (FWHM) of 5 mm, then quantified brain activation by calculating the amplitude of low-frequency fluctuation (ALFF) (*Zang et al., 2007*). We used the program *3dRSFC*, which is a component of Analysis of Functional NeuroImages (AFNI) (*Cox, 1996*; *Cox and Hyde, 1997*), to compute ALFF in voxel space. ALFF was computed from the 0.01–0.08 Hz frequency band, within a gray matter-only brain mask. Computed voxel space ALFF values were normalized to the global mean of each individual subject. Statistical analyses were performed in voxel space using the Nistats Python library. We used a significance threshold of $p < 0.05$ and a minimum cluster size of 12 voxels (~100 mm$^3$) and controlled for multiple comparisons using FDR.

### Implications of T2DM duration

To evaluate the implications of T2DM chronicity, we analyzed whole-brain gray matter volume with time since T2DM diagnosis as a regressor. Time since diagnosis was derived from self-reported age at T2DM diagnosis. To improve the accuracy of self-reported values, we averaged the reported age of onset values from three visits, separated by multiple years. To estimate the degree to which T2DM progression was associated with gray matter loss relative to age, we calculated the ratio of regression coefficients corresponding to T2DM duration and age, and expressed it as a percentage, using Fieller's theorem (*Fieller, 1954*) to quantify the confidence interval (95%) for this ratio.

### Implications of metformin treatment

For patients with T2DM, we evaluated whether metformin, a first-line medication for the treatment of T2DM, was associated with improved outcomes in terms of cognition, atrophy, and/or brain activation. To isolate medication effects specific to metformin, we compared subjects who reported not taking any medications to treat T2DM to subjects who reported taking metformin but no other medications. For these comparisons, we exact-matched for sex, education, and hypertension, and coarse-matched for age (bin size of 5 years) and disease duration (bin size of 3 years). Since UK Biobank did not measure HbA1c levels, we also included body mass index (BMI) as a regressor since it was the only available proxy measure for disease severity (*Bower et al., 2017*; *Bae et al., 2016*).

### Implications of sex

To determine whether results detected in the sample might be driven by sex-specific factors, we additionally performed analyses separately in females and males. For these, we evaluated our cognitive and neurobiological measures in association with age and T2DM and quantified the overlap separately for the two subsamples.

## Meta-analysis of published literature (N = 24,185)

### Search strategy and selection criteria (cognition)

We conducted a literature search for peer-reviewed articles published up to 28 August 2020 from PubMed/Medline using the following search terms: "type-2-diabetes," "diabetes mellitus, type 2," "insulin-resistance," < AND > "cognition," "cognitive-function," "cognitive-dysfunction," "cognitive-performance," and "neuropsychological tests." Search results were filtered to include manuscripts that had undergone peer-review, were published in English with full-text availability, and reported relevant results. Our cognitive meta-analysis adhered to PRISMA guidelines (*Page et al., 2021*).

We included studies that compared cognitive performance between people diagnosed with T2DM and HCs. We excluded studies that (1) included participants with neurological or psychiatric diagnoses, (2) utilized treatment interventions without first obtaining baseline cognitive measurements, (3) included only diagnostic threshold instruments for dementia (e.g., the Mini-Mental State Examination [MMSE]), (4) included a novel cognitive test without adequate explanation of the scoring procedures, (5) did not perform age- and education-matching of the participants diagnosed with T2DM to their HC, or (6) failed to provide summary statistics needed to calculate effect sizes. In the latter case, the authors were contacted to obtain relevant data.

Our literature search yielded 219 articles; relevant reviews were also screened for eligible studies. Seventy-five articles were identified for full-text evaluation; 34 studies were eligible for inclusion. Among the studies that were excluded (*Supplementary file 2*), 8 featured inadequate testing or scoring procedures, 14 included secondary analyses of the same patient sample that was used in previous publications, and 5 failed to perform appropriate education-matching of the study groups. Furthermore, one longitudinal study did not report baseline scores and another reported inconsistent sample sizes. Fifteen authors were contacted to obtain data not provided in the text; three authors provided the data requested, and the remaining 12 studies were excluded. Eligible studies included a total of 4735 subjects diagnosed with T2DM and 17,496 HC (*Supplementary file 3*).

## Data analysis (cognition)

We extracted data, including publication year, authors, sample demographics, and cognition, from all included studies. We extracted baseline data only from longitudinal studies to avoid practice effects. We sorted individual cognitive tests into several domains, including *abstract reasoning*, *executive function*, *processing speed*, *numeric memory*, *visual memory*, *verbal memory*, *verbal fluency*, *visuospatial reasoning*, and *working memory* (*Supplementary file 4*).

Statistical analyses were performed using R version 3.6.1 (*R Development Core Team, 2018*) and the Metafor package version 2.4-0 (*Viechtbauer, 2010*). Cognitive differences between participants diagnosed with T2DM and HC were determined by calculating standardized mean difference (SMD) effect sizes and 95% CIs for all cognitive domains. Effect size analyses were chosen to account for within-domain variability in the type and sensitivity of cognitive tests across different reports. We calculated effect sizes as Cohen's d by dividing the mean difference in group scores by the pooled standard deviation of individual domains (*Cohen, 1988*); an SMD (Cohen's d) of –1.0 was interpreted as a difference of 1 SD in the negative direction. We used random-effects models to account for variability between samples not due to sampling error with significance at p<0.05, and effect size heterogeneity was evaluated using values for Cochran's Q and I (*Moheet et al., 2015*; *Higgins et al., 2003*). Publication bias was evaluated with funnel plots. We applied Bonferroni correction to account for multiple comparisons across cognitive domains.

## Search strategy and selection criteria (brain)

We used NeuroQuery (*Dockès et al., 2020*) to conduct a meta-analysis of published neurobiological results associated with T2DM and age. NeuroQuery is an automated Coordinate-Based Meta-Analysis (*Fox et al., 1998*; *Van Horn et al., 2004*; *Yarkoni et al., 2010*) tool based on a database of z-scores collected by crawling through texts and tables of published research articles by an automated algorithm (*Yarkoni et al., 2011*). We utilized NeuroQuery to address limitations of standard approaches to meta-analyses of neuroimaging results, which rely on summary statistics and thus risk overfitting to what typically comprise a relatively small number of in-sample studies (i.e., they fail to generalize to out-of-sample studies). NeuroQuery optimizes for rigor and reproducibility by utilizing predictive modeling, a higher threshold for results than statistical significance. In a quantitative evaluation of its generalization performance with 16-fold cross-validation and 10:90 test-train splits, NeuroQuery was found to accurately produce brain maps for out-of-sample neuroimaging studies (*Yarkoni et al., 2011*). We note that because of NeuroQuery's criteria for neuroimaging data quality and completeness in reporting, the algorithm draws only from journals that focus on functional neuroimaging results, and thus can exclude some general interest and non-neurological medical journals. This exclusion criterion is important to reduce false positives and ensure the quality and relevance of the compiled results. The database in total contains 149,000 neuroimaging papers and represents the single largest database of neuroimaging foci to date. By the law of large numbers, NeuroQuery therefore provides the

most unbiased approach to choosing representative papers, even at the risk of excluding relevant and well-cited articles specific to any one field.

Using the collected database of articles, NeuroQuery applies a multivariate model to predict the spatial distribution of voxel activations corresponding to a search term. The search terms we used to obtain the meta-analytic maps were "diabetic" and "age." These terms identified the 30 most relevant neuroimaging studies for T2DM and 30 most relevant neuroimaging studies for age (*Supplementary file 5*). To account for any errors in the automated search results, the identified set of studies were cross-validated by an independent manual search using the same search terms for Google Scholar and PubMed to verify their relevance, as well as to confirm that they included T2DM age-matched HC and T2DM (not type 1 diabetes). In the T2DM datasets, 23 were fMRI (ALFF), 2 were structural (T1), 3 were FDG positron emission tomography (PET), and 2 were tractography (diffusion tensor imaging [DTI]). In the age datasets, 22 were fMRI (ALFF), 3 were structural (T1), and 5 were tractography (DTI, diffusion weighted imaging).

## Data analysis (brain)

For region- and voxel-level comparisons of the meta-analytic T2DM and age maps from NeuroQuery with their structural and functional counterparts from UK Biobank, the meta-analytic statistical maps were transformed onto comparable coordinate space and spatial resolution. At the voxel level, the meta-analytic maps were resampled to the standard MNI affine (the transformation matrix that maps from voxel indices of the data array to actual real-world locations of the brain; no registration was required as images were already aligned). For region-level comparisons, the transformed voxel maps were coarse-grained to the 45 regions of interest from UK Biobank by masking with each individual region and computing the mean activation of the masked voxels as the representative region value.

# Results

## Cognitive correlates with age and T2DM

Individuals without T2DM showed age-based cognitive effects across all domains in the UK Biobank (*Figure 1A*). The strongest effects were observed in *executive function*, which showed 1.9% ± 0.1% decrease in performance per year (N = 2450, T = –17.2, p<1e–10), and *processing speed*, which showed 1.5% ± 0.2% decrease in performance per year (N = 2525, T = –22.8, p<1e–10). Our analyses identified further cognitive deficits associated with T2DM that were consistent with accelerated age-related cognitive decline (*Figure 1B*). As with aging, the strongest T2DM effects were also observed in *executive function*, which showed a further 13.1% ± 6.9% decrease in performance, beyond age-related effects (T2DM: N = 446; HC: N = 446; T = –3.7, p=0.001), and *processing speed*, which showed a further 6.7% ± 3.2% decrease in performance, beyond age-related effects (T2DM: N = 454; HC: N = 454; T = –4.1, p=0.0002). A more modest decline (3.7% ± 2.3%) was observed in *numeric memory* (~2–3 s) (T2DM: N = 483; HC: N = 483; T = –3.2, p=0.007), whereas *abstract reasoning* (T2DM: N = 886; HC: N = 886; T = –2.4, p=0.08) and *reaction time* (T2DM: N = 914; HC: N = 914; T = –1.0, p=0.32) were not statistically significant. Our meta-analysis confirmed that individuals with T2DM exhibited markedly lower performance when compared to age- and education-matched controls, over an even broader set of domains (*Figure 1C*). These again included *executive function* (K = 18, d = –0.40, p=0.009), *processing speed* (K = 31, d = –0.34, P = 5e–8), and *numeric memory* (~2–3 s) (K = 16, d = –0.21, p=0.05), as well as *abstract reasoning* (K = 8, d = –0.36, p=1e–7), *immediate (~30 s) verbal memory* (K = 23, d = –0.39, p=0.001), *delayed (~20 min) verbal memory* (K = 21, d = –0.21, p=0.005), *verbal fluency* (K = 25, d = –0.37, p=2e–8), *visuospatial reasoning* (K = 13, d = –0.32, p=4e–7), and *working memory* (K = 12, d = –0.36, p=0.002) (*Supplementary file 6*).

## Neurobiological correlates with age and T2DM

### Brain atrophy

HCs (N = 4,775) showed a linear decrease in brain gray matter with age. This was most pronounced in the *ventral striatum*, which showed a 1.0% ± 0.06% decrease per year (T = –30.4, p<1e–10), and *Heschl's gyrus*, which also showed a 0.9% ± 0.06% decrease per year (T = –30.4, p<1e–10) (*Figure 2A*). Compared to their age-matched HC, T2DM patients showed further decreases in gray matter beyond typical age-related effects (T2DM: N = 821; HC: N = 821). These included both cortical and subcortical

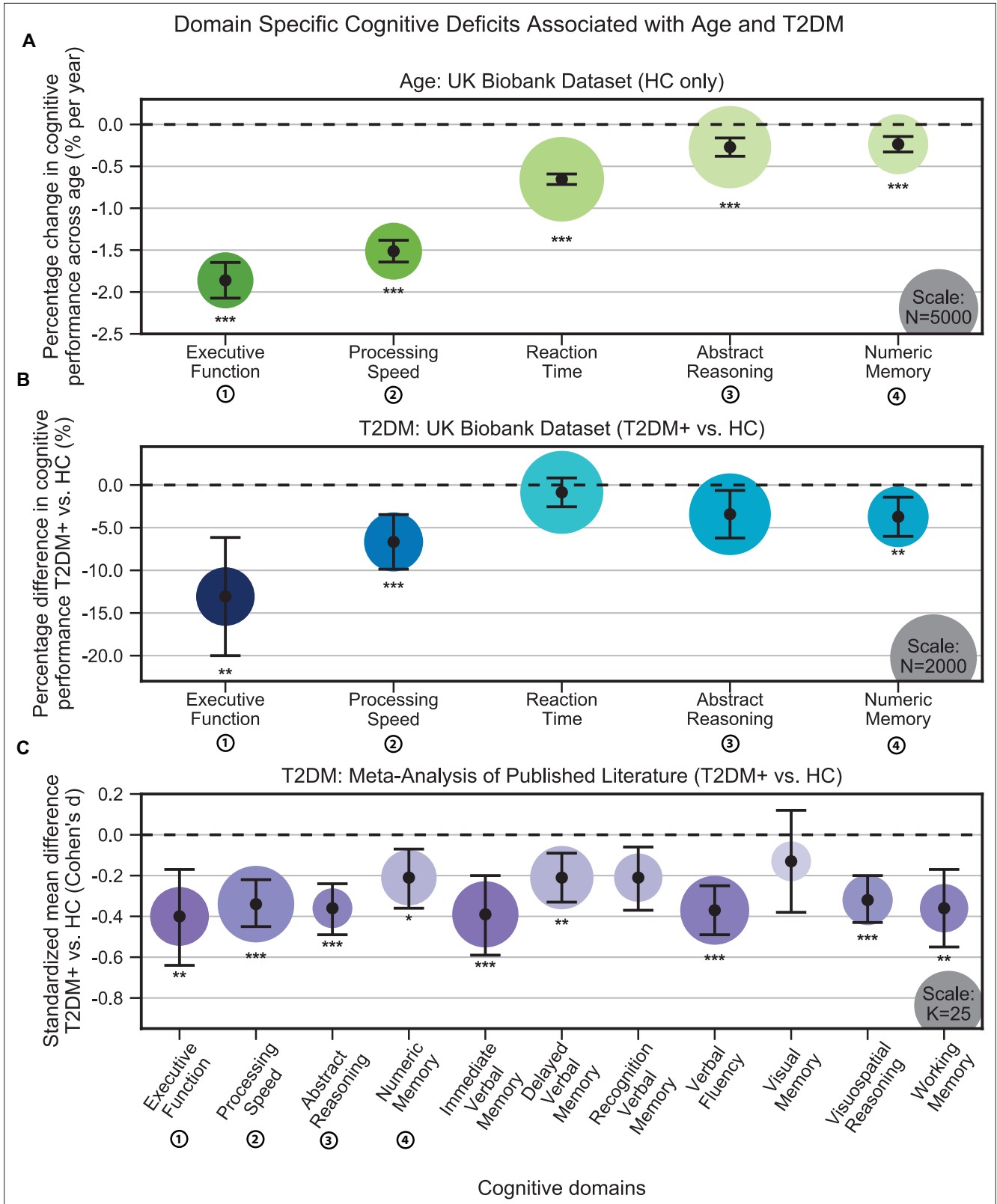

**Figure 1.** Cognitive deficits are apparent with respect to both age and type 2 diabetes mellitus (T2DM) diagnosis. (**A**) Using the UK Biobank dataset, we performed a quantitative analysis of the effects related to age on cognitive performance across five cognitive domains. Associated changes were derived from estimated regression coefficients as percentages and are shown on the y-axis. Age was associated with significant deficits in all five domains, with the strongest effects observed in *executive function* and *processing speed*. (**B**) Using the same dataset, we also analyzed cognitive

*Figure 1 continued on next page*

*Figure 1 continued*

performance in T2DM, with negative values on the y-axis representing performance below that of age-, sex-, and education-matched healthy controls (HCs). As per age effects, *executive function* and *processing speed* showed the highest magnitude changes. (**C**) Cognitive deficits identified in UK Biobank data were confirmed by our meta-analysis, which included 11 domains from 34 studies. Average effect sizes (Cohen's d) corresponding to T2DM are shown on the y-axis. Values below the cutoff line (y = 0) indicate cases in which subjects with T2DM performed less well than age- and education-matched HC. Numbers next to labels identify domains common across panels. Marker sizes represent sample sizes scaled (per area) as indicated in the bottom-right corner of each panel. On panel (**C**), sample size indicates the number of individual studies. Underlying sample size distributions can be found in *Appendix 1—figures 1C and 2C*. Error bars are 95% CI. *p≤0.05; **p≤0.01; ***p≤0.001, Bonferroni corrected.

The online version of this article includes the following figure supplement(s) for figure 1:

**Figure supplement 1.** Cognitive differences associated with sex in the UK Biobank dataset across the five cognitive domains.

**Figure supplement 2.** Cognitive differences associated with age (**A**) and type 2 diabetes mellitus (T2DM) (**B**) in the UK Biobank dataset across the five cognitive domains, analyzed separately within males and females.

**Figure supplement 3.** Treatment of type 2 diabetes mellitus (T2DM) patients with metformin had no impact on cognitive deficits.

regions, with the most severe atrophy observed in the *ventral striatum*, which showed on average a 6.2% ± 1.6% further decrease in volume, beyond age-related effects (T = –7.5, p<1e-10), in the *cerebellum* with an additional 4.9% ± 1.1% decrease in volume, beyond age-related effects (T = –8.8, p<1e-10), and in the *putamen*, which showed a 4.7% ± 2.3% further decrease, beyond age-related effects (T = –4.1, p=0.002) (*Figure 2B*).

### Brain activity

Age was associated with functional reorganization of brain activation (ALFF), rather than global decrease or increase. Brain activation in T2DM showed similar reorganization. Normalized to whole-brain activation, both age (HC: N = 2250) and T2DM (T2DM: N = 646, HC: N = 646) were associated with decreased activation in the *caudate and premotor cortex*, and with increased brain activity in the *subgenual area, orbitofrontal cortex, posterior cingulate cortex,* and *brainstem* (*Figure 3A*).

### NeuroQuery

Our meta-analysis of 60 multimodal neuroimaging studies (30 age-specific, 30 T2DM-specific) independently identified the same regions as UK Biobank (*premotor cortex, caudate, posterior cingulate gyrus*), but additionally identified clusters of decreased activity in *Broca area* and the *frontal eye fields* and increased activity in the *thalamus* and *inferior temporal gyrus* (*Figure 3B*).

## Neurocognitive changes associated with T2DM and age overlap, consistent with common pathways

Together, these analyses confirm that T2DM patients show evidence of neurocognitive deficits, with the most consistent and profound effects observed in structural atrophy. Even after controlling for education, cognitive deficits remained statistically significant. Both age and T2DM implicated the same areas of greatest vulnerability: for brain atrophy, these were the *ventral striatum, Heschl's gyrus,* and *cerebellum*; for cognition, these were *executive function* and *processing speed*. When assessed across all brain regions, T2DM-related patterns in brain atrophy exhibited strong overlap with those associated with age (r = 0.60, p=0.0002) (*Figure 2—figure supplements 1 and 2A*). Similarly, T2DM-related changes in brain activation (ALFF) also exhibited significant overlap with those associated with age (r = 0.64, p=0.00004) (*Figure 3—figure supplements 1 and 2B*). The meta-analysis, which included multimodal neuroimaging measures (not only atrophy and brain activity, but also glucose uptake via FDG-PET), also yielded equivalent results in terms of the overlap between neurobiological effects of T2DM and age (r = 0.58, p=0.0005) (*Figure 2—figure supplements 1 and 2C*).

## T2DM chronicity exacerbates neurocognitive symptoms

Neurocognitive effects were more severe with increased disease duration, particularly for structural changes (T = –3.8, p=0.0002) (*Figure 4*). T2DM progression was associated with 26% ± 14% acceleration of typical neurogenerative age-related effects as per the linear shift along the horizontal time axis shown in *Figure 4*.

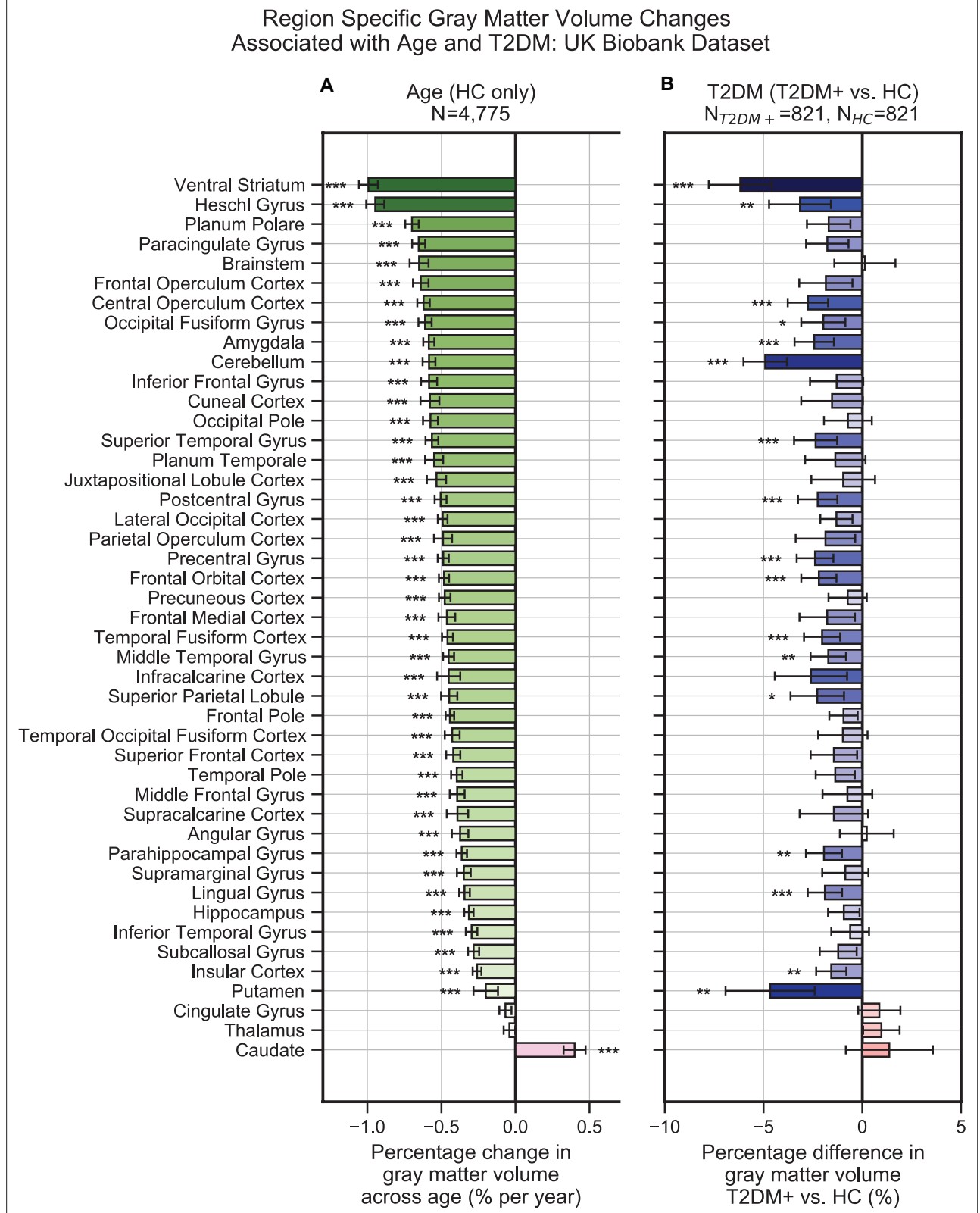

**Figure 2.** Widespread gray matter atrophy can be observed with respect to both age and type 2 diabetes mellitus (T2DM) diagnosis status. Using the UK Biobank dataset, we measured gray matter atrophy across 45 anatomical regions. Associated changes were derived from estimated regression coefficients as percentages and are shown on the x-axes. (**A**) We observed significantly decreased gray matter volume in both cortical and subcortical brain regions with respect to age in healthy controls (HCs). Age was associated with an average of ~0.5% brain-wide decrease in gray matter volume per

*Figure 2 continued on next page*

*Figure 2 continued*

year, most prominently for the *ventral striatum* and *Heschl's gyrus*. (**B**) Gray matter atrophy was also seen in patients diagnosed with T2DM compared to age-matched HC, most prominently for the *ventral striatum*, *cerebellum*, and *putamen*. The distribution of T2DM-related effects overlapped with those associated with age, with degeneration of the *ventral striatum* and preservation of the *thalamus* and *caudate*. Underlying sample size distributions can be found in *Appendix 1—figures 1A and 2A*. Error bars are 95% CI. *p≤0.05; **p≤0.01; ***p≤0.001, Bonferroni corrected.

The online version of this article includes the following figure supplement(s) for figure 2:

**Figure supplement 1.** Effects of age and type 2 diabetes mellitus (T2DM) exhibited strong correlations within datasets and modalities.

**Figure supplement 2.** Scatterplots corresponding to the statistically significant cells in *Figure 2—figure supplement 1*.

**Figure supplement 3.** Gray matter volume (normalized for head size) differences associated with sex in the UK Biobank dataset across the 45 anatomical regions.

**Figure supplement 4.** Plots representing trends in total gray matter volume (normalized for head size) across age, sex, and type 2 diabetes mellitus (T2DM).

**Figure supplement 5.** Region-specific gray matter volume deficits associated with age (**A**) and type 2 diabetes mellitus (T2DM) (**B**) in the UK Biobank dataset, analyzed separately within females and males.

**Figure supplement 6.** We quantified correlations (Pearson's *r*) among the region-specific gray matter volume deficits associated with age and type 2 diabetes mellitus (T2DM), which we quantified in the UK Biobank dataset separately for females and males.

**Figure supplement 7.** Scatterplots corresponding to the most relevant cells in *Figure 2—figure supplement 6*.

**Figure supplement 8.** Treatment of type 2 diabetes mellitus (T2DM) patients with metformin had no impact on gray matter atrophy.

## T2DM patients treated with metformin do not demonstrate improved neurocognitive symptoms

Even after matching groups for disease duration and controlling for BMI, T2DM patients who were treated with metformin alone (N = 498) did not differ with respect to cognition or brain atrophy compared to T2DM patients who were unmedicated (N = 352) (*Figure 1—figure supplement 3*, *Figure 2—figure supplement 8*). Likewise, treatment status showed no significant impact on resting-state brain activity.

## Age- and T2DM-associated effects were consistent in females and males

For the UK Biobank dataset, we identified marked sex-related differences both in neurobiological and cognitive measures, consistent with the literature (*Figure 1—figure supplement 1*, *Figure 2—figure supplement 3*, *Figure 3—figure supplement 3*; *Sanchis-Segura et al., 2019*; *Jain et al., 2015*). All reported age and T2DM effects were seen for both males and females, and strongly correlated between them (*Figure 2—figure supplement 6*, *Figure 3—figure supplement 5*). However, age and T2DM effects were stronger in males (*Figure 1—figure supplement 2*, *Figure 2—figure supplement 5*, *Figure 3—figure supplement 4*) and neurodegeneration (brain atrophy, in particular) was more similar across sex for age ($r_{age}$ = 0.93, Bonferroni-corrected p≤0.001) than for T2DM ($r_{T2DM}$ = 0.51, Bonferroni-corrected p≤0.01). Consistent with the latter, the overlap between effects associated with age and T2DM was statistically significant in males (gray matter volume: *r* = 0.77, p=4e–9; brain activation [ALFF]: *r* = 0.51, p=0.002) but not in the female-only subsample (*Figure 2—figure supplement 6*, *Figure 2—figure supplement 7*, *Figure 3—figure supplement 5*, *Figure 3—figure supplement 6*). All females in our UK Biobank sample were menopausal and not on hormone replacement therapy. For our meta-analyses, we were unable to perform a sex-based comparison as the underlying articles from which our data were derived did not always perform sex-matching across their subject pools and did not provide access to the individual subject-level data that would be required to control for sex in our analyses.

## Discussion

The UK Biobank dataset confirms that T2DM patients show deficits in cognitive performance compared to HC, even after controlling for age, sex, education, and hypertension. These findings were supported by meta-analysis of the published literature. Deficits in cognitive performance were accompanied by marked brain atrophy in the T2DM sample compared to age-matched HC. The

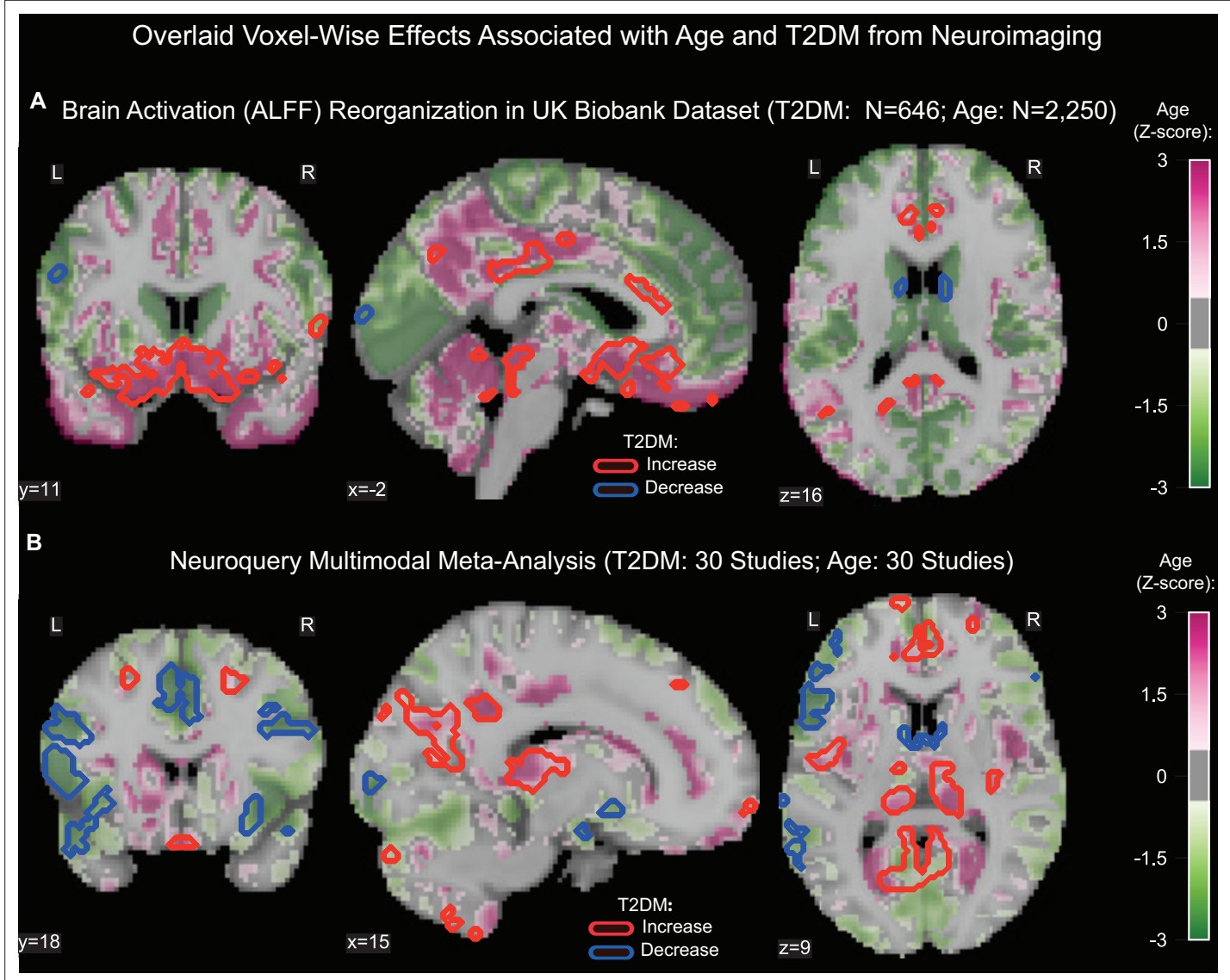

**Figure 3.** Age- and type 2 diabetes mellitus (T2DM)-associated reorganization patterns in brain activity are significant and overlap. (**A**) For functional MRI data obtained from the UK Biobank dataset, we used the amplitude of low-frequency fluctuation (ALFF) to quantify brain activation. Effects linked to age are shown in the form of an unthresholded z-map represented by the pink-green color gradient, with pink indicating increased activation and green showing decreased. T2DM-related effects were thresholded (minimum cluster size ~100 mm³, false discovery rate [FDR] p<0.05) to result in significant clusters. The outlines of these significant clusters are overlaid on the age-related z-map to demonstrate overlapping effects. The largest significant clusters with respect to T2DM were in the *subgenual area* (increased) *orbitofrontal cortex* (increased), *premotor cortex* (decreased), and *caudate* (decreased). The highlighted regions were similarly impacted across age, indicating substantial overlap between the two contrasts. Underlying sample size distributions can be found in *Appendix 1—figures 1B and 2B*. (**B**) Using multimodal neuroimaging data, we performed a meta-analysis for the same contrasts using NeuroQuery. We extracted contrast maps for age and T2DM with NeuroQuery and overlaid the outlines of thresholded (minimum cluster size ~100 mm³, FDR p<0.05) z-maps from T2DM on unthresholded z-maps belonging to age. The overlapping effects were present in several regions, most importantly in the *posterior cingulate gyrus, thalamus, caudate,* and *premotor cortex*. These results support the hypothesis that neurodegeneration in both T2DM and aging may share common mechanistic pathways.

The online version of this article includes the following figure supplement(s) for figure 3:

**Figure supplement 1.** Effects of age and type 2 diabetes mellitus (T2DM) exhibited strong correlations within datasets and modalities.

**Figure supplement 2.** Scatterplots corresponding to the statistically significant cells in *Figure 3—figure supplement 1*.

**Figure supplement 3.** Resting-state brain activation (amplitude of low-frequency fluctuation [ALFF]) differences associated with sex in the UK Biobank dataset.

*Figure 3 continued on next page*

*Figure 3 continued*

**Figure supplement 4.** Age and type 2 diabetes mellitus (T2DM)-associated reorganization patterns in brain activation (amplitude of low-frequency fluctuation [ALFF]) were overlaid separately for females (**A**) and males (**B**).

**Figure supplement 5.** We quantified correlations (Pearson's *r*) among the region-specific changes in brain activation (amplitude of low-frequency fluctuation [ALFF]) patterns associated with age and type 2 diabetes mellitus (T2DM), which we quantified in the UK Biobank dataset separately for females and males.

**Figure supplement 6.** Scatterplots corresponding to the primary cells in *Figure 3—figure supplement 5*.

atrophy was most severe (6.2% gray matter loss compared to HC) in the *ventral striatum*, a region critical to learning, decision-making, goal-directed behavior, and cognitive control. These cognitive functions, collectively known as *executive functioning*, were (with *processing speed*) also those most affected by T2DM. Neurodegeneration severity for all regions increased with longer disease duration. We detected qualitatively consistent results in females and males; however, males exhibited stronger effects in relation to T2DM (*Figure 2—figure supplement 4*). This result is consistent with the well-established neuroprotective effects of female hormones such as estrogen (*Behl, 2002*). This result also suggests that the T2DM neurological effects observed result from chronic degenerative processes, which, for our female participants, may have been at least partially ameliorated prior to menopause.

Our findings indicate that structural brain imaging, in particular, can provide a clinically valuable metric for identifying and monitoring neurocognitive effects associated with T2DM. Normalizing across sample sizes to compare the measures of neurocognitive effects – *structural MRI*, *functional MRI*, and *cognitive testing* – structural atrophy showed global effects that were far more statistically robust

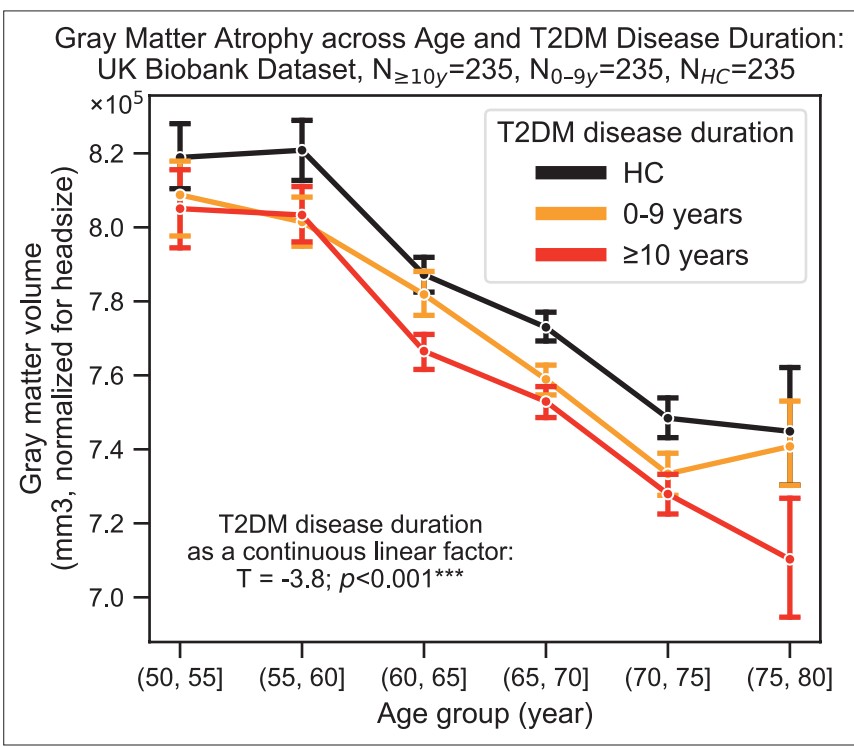

**Figure 4.** Progression of type 2 diabetes mellitus (T2DM) disease is significantly associated with gray matter atrophy, accelerating neurodegenerative effects seen in brain aging. For a quantitative evaluation of the impact of T2DM progression on gray matter volume, we considered time since T2DM diagnosis as the main factor of interest from the UK Biobank dataset. The T2DM+ cohort was divided into two groups based on disease duration (separated at 10 years) with a healthy control (HC) cohort also included for visualization purposes. We matched age, sex, education, and hypertension across these three groups and performed linear regression within T2DM+ subjects focusing on disease duration. Evaluation of our sample suggested that time since diagnosis was a significant factor, with each year after diagnosis of T2DM associated with an additional ~0.26 ± 0.14 years of brain aging beyond that of age-matched HC. Underlying sample size distributions can be found in *Appendix 1—figure 3*. Error bars are standard error of the mean. *p≤0.05, **p≤0.01, ***p≤ 0.001.

(p<2e–10) than either global cognitive measures (p=0.0001) or global brain activation (p=0.002). One additional advantage of structural MRI over cognitive testing is that the former avoids confounding associated with education and practice effects, and therefore may be more interpretable in real-world clinical settings in which matching for education and practice effects is not feasible. Structural MRI also showed advantages as a biomarker over a functional MRI-derived measure of brain activation (ALFF). The reorganization of brain activity seen with T2DM may reflect the brain's switch to less metabolically expensive networks to conserve energy in the face of diminishing access to glucose, a pattern previously documented in aging (*Tomasi et al., 2013*; *Weistuch et al., 2021*; *Tomasi and Volkow, 2012*; *Mujica-Parodi et al., 2020*). Yet activation patterns that are spatially reorganized, rather than globally increased or decreased, are less straightforward to quantify. Moreover, functional MRI is an inherently more complex measure than structural MRI, reflecting both neuronal and hemodynamic influences. Each of these influences may be differentially affected by T2DM, further complicating its interpretation in a clinical setting.

The localization of brain atrophy in T2DM to the *ventral striatum*, followed by the *cerebellum*, may reflect the fact that these two brain regions contain the densest concentrations of insulin-dependent GLUT-4 (*El Messari et al., 2002*; *Kobayashi et al., 1996*; *El Messari et al., 1998*; *Vannucci et al., 1998*) compared to non-insulin-dependent isoforms GLUT-1 and GLUT-3. The *ventral striatum* functions as a critical hub within the reward circuit, integrating inputs (including external stimuli) from both cortical and subcortical regions, and therefore is a key structure required for all learning. Rat studies have shown modulation of nitric oxide within the *ventral striatum* to control release of acetylcholine (*Prast et al., 1995*), a neurotransmitter severely reduced in dementia (*Kurotani et al., 2003*) and a target for its pharmaceutical treatment (*Ferreira-Vieira et al., 2016*; *Muramatsu et al., 2019*). Release of nitric oxide is insulin dependent and reduced in T2DM (*Tessari et al., 2010*). Together, these suggest a potential mechanistic pathway between insulin resistance, atrophy of the *ventral striatum*, and widespread deficits with respect to learning. In this context, memory deficits may be primarily driven by failure to *encode* rather than failure to *retrieve*, which would be consistent with our results that did not identify the hippocampus to be one of the regions most affected. Importantly, the structural and functional changes associated with T2DM show marked overlap with age-related effects but appear earlier. This, on the one hand, suggests that neurocognitive changes seen in T2DM may progress via a common mechanistic trajectory as normal brain aging, but which is accelerated. From another perspective, the overlap implies that brain aging itself may be a metabolic syndrome driven by impaired brain insulin signaling and glucose metabolism, the same processes that, outside the brain, are well-established with respect to T2DM.

Our analyses had two limitations, inherent in the datasets analyzed, which represent important directions for future research. First, our use of a lifespan dataset permitted tracking how variables change with age, but not for the same subjects. A more rigorous assessment of phase shift between trajectories of neurodegeneration for patients with T2DM and HC would be made possible only with a longitudinal study. Second, while we had access to disease duration and BMI, we did not have HbA1c measures, which would have provided a more direct measure of disease severity. While metformin was not found to be associated with better neurocognitive measures, even when matched to unmedicated patients with equivalent disease duration and after controlling for BMI (a proxy measure for disease severity; *Bower et al., 2017*; *Bae et al., 2016*), it was not possible to determine other diabetes-related characteristics. As such, our medication findings should be considered suggestive but not conclusive.

Consistent with findings from earlier studies that focused on the brain and energy metabolism (*Sokoloff, 1955*; *Clark, 1999*), our results suggest that T2DM and its progression may accelerate pathways associated with typical brain aging. As T2DM diminishes s glucose availability within the brain, this chronic loss of energy may compromise the brain's structure and function . We consider the possibility that, by the time T2DM is formally diagnosed, neuronal insulin resistance may have already resulted in significant damage. As such, our findings underscore the need for additional research into brain-based biomarkers for T2DM and treatment strategies that specifically target its neurocognitive effects (*Kaiser et al., 2018*).

## Additional information

### Competing interests
Deborah J Wexler: is part of a Novo Nordisk data monitoring committee service for semaglutide in SOUL and FLOW trials. The author has no other competing interests to declare. Bradford Dickerson: received royalties from Cambridge University Press and Oxford University Press, and consulting fees from Acadia, Alector, Arkuda, Biogen, Denali, Lilly, Merck, Novartis, Takeda, Wave LifeSciences (unrelated to the present work). Also participates in a Lilly Data Safety Monitoring Board (unrelated to the present work) and participates in leadership roles in Alzheimer's Association and Association for Frontotemporal Degeneration. The author has no other competing interests to declare. Eva-Maria Ratai: received honoraria from Harvard Catalyst. The author has no other competing interests to declare. The other authors declare that no competing interests exist.

### Funding

| Funder | Grant reference number | Author |
| --- | --- | --- |
| W. M. Keck Foundation | | Lilianne R Mujica-Parodi |
| National Science Foundation | NSFNCS-FR 1926781 | Lilianne R Mujica-Parodi |
| Baszucki Brain Research Fund | | Lilianne R Mujica-Parodi |

The funders had no role in study design, data collection and interpretation, or the decision to submit the work for publication.

### Author contributions
Botond Antal, Formal analysis, Investigation, Methodology, Visualization, Writing – original draft, Writing – review and editing; Liam P McMahon, Data curation, Formal analysis, Investigation, Methodology, Writing – original draft, Writing – review and editing; Syed Fahad Sultan, Formal analysis, Investigation, Methodology, Writing – original draft, Writing – review and editing; Andrew Lithen, Investigation; Deborah J Wexler, Bradford Dickerson, Eva-Maria Ratai, Writing – review and editing; Lilianne R Mujica-Parodi, Conceptualization, Data curation, Funding acquisition, Methodology, Project administration, Resources, Supervision, Visualization, Writing – original draft, Writing – review and editing

### Author ORCIDs
Botond Antal  http://orcid.org/0000-0002-0775-6033
Lilianne R Mujica-Parodi  http://orcid.org/0000-0002-3752-5519

### Decision letter and Author response
Decision letter https://doi.org/10.7554/eLife.73138.sa1
Author response https://doi.org/10.7554/eLife.73138.sa2

## Additional files

### Supplementary files
• Supplementary file 1. Summary of all relevant UK Biobank datafields.

• Supplementary file 2. List and justification for studies excluded from our cognitive meta-analysis.

• Supplementary file 3. Characteristics of patients who underwent cognitive testing in studies included in our meta-analysis.

• Supplementary file 4. Summary of cognitive functions assessed, with corresponding instruments.

• Supplementary file 5. Studies identified as most relevant for each key word by NeuroQuery algorithm.

• Supplementary file 6. Study estimates of cognitive meta-analysis.

• Transparent reporting form

- Reporting standard 1. Strobe checklist.
- Reporting standard 2. Prisma checklist.

## Data availability

All analyses reported in this manuscript were on either publicly available data (UK Biobank) or meta-analyses of listed published articles. Source code for all analyses conducted for this manuscript is uploaded as Source Code (https://github.com/lcneuro/pub_t2dm_age_meta, copy archived at swh:1:rev:81c32d4b6c25191eef4028a7dd2337b71af59628).

The following previously published dataset was used:

| Author(s) | Year | Dataset title | Dataset URL | Database and Identifier |
|---|---|---|---|---|
| Sudlow C, Gallacher J, Allen N | 2015 | UK Biobank | https://www.ukbiobank.ac.uk/ | UK Biobank, Biobank |

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

## Appendix 1

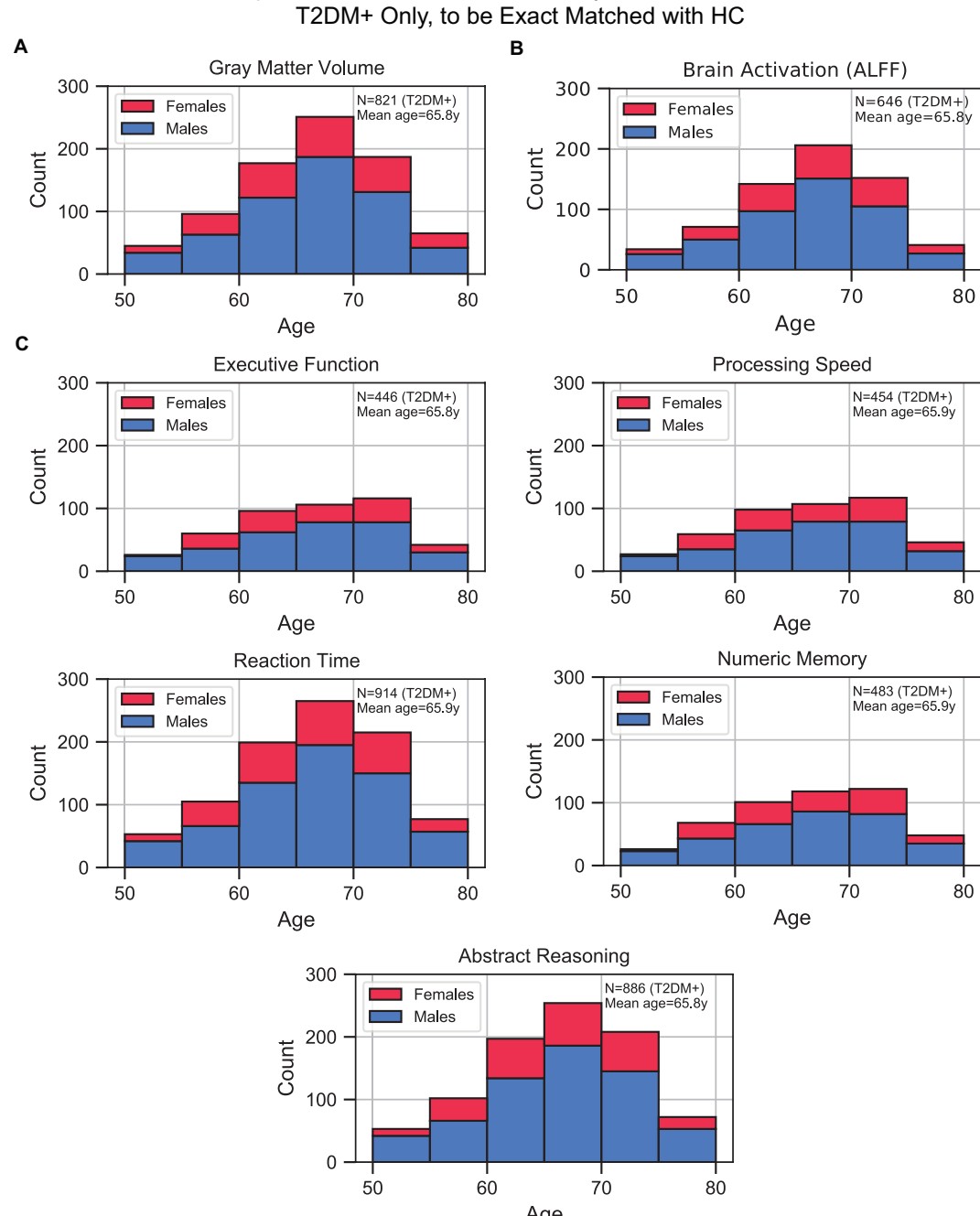

**Appendix 1—figure 1.** Sample size distributions in type 2 diabetes mellitus (T2DM) vs. healthy control (HC) analyses. The values shown represent T2DM+ only, T2DM+ were matched with an equivalent number of HCs, resulting in twice the number of samples in every analysis. The bars are stacked. (**A**) Gray matter volume analyses, (**B**) brain activation (amplitude of low-frequency fluctuation [ALFF]) analyses, and (**C**) cognition analyses (five domains).

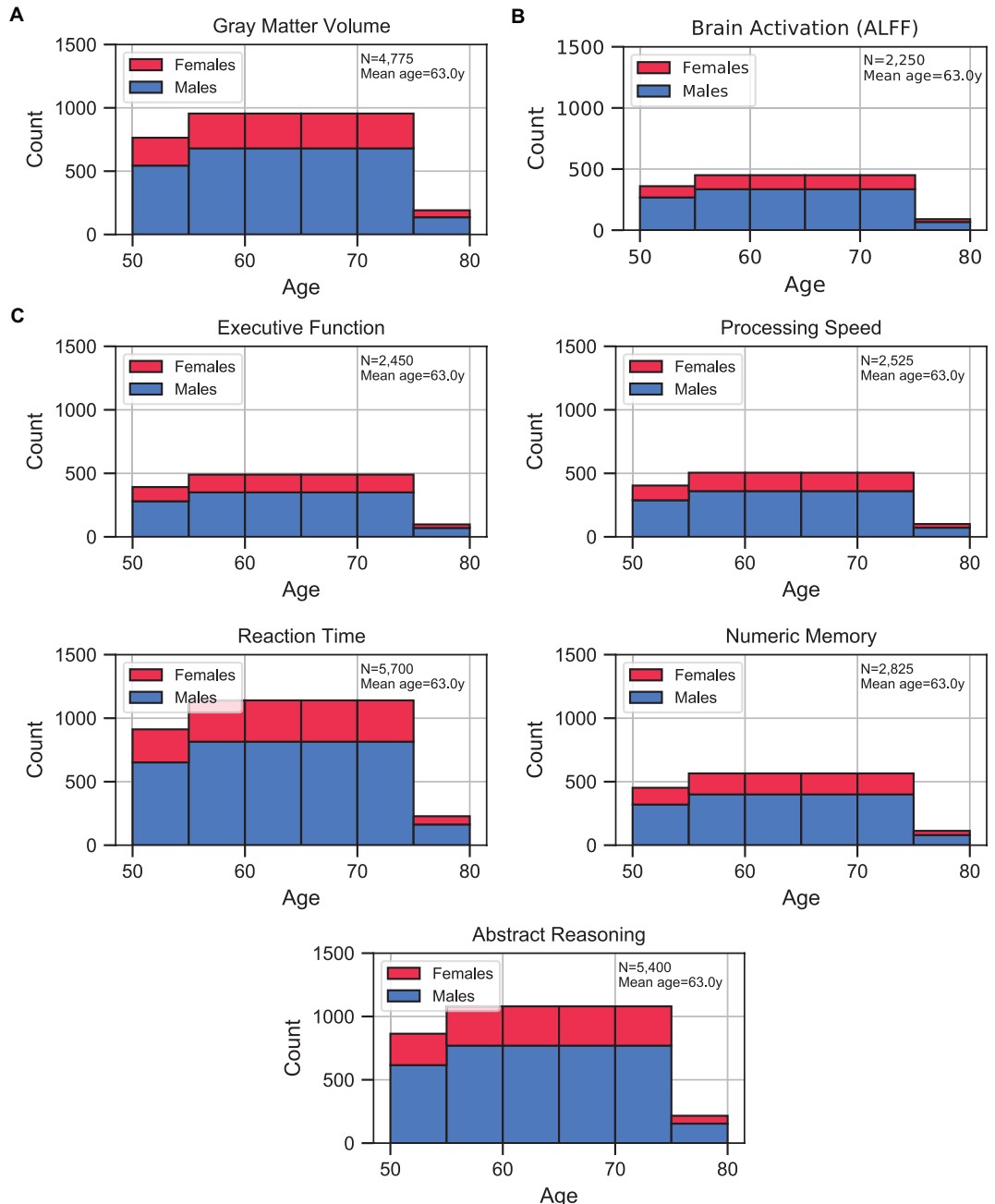

**Appendix 1—figure 2.** Sample size distributions of the analyses that investigated age as the factor of interest. Samples included healthy control (HC) only and were matched for sex, education, and hypertension across age. The matching method resulted in 'sample chains' across age: within each chain, samples were equivalent in terms of sex, education, and hypertension. The bars are stacked. (**A**) Gray matter volume analyses, (**B**) brain activation (amplitude of low-frequency fluctuation [ALFF]) analyses, and (**C**) cognition analyses (five domains).

## Sample Sizes for the Analysis of Gray Matter Volume with Respect to T2DM Duration, UK Biobank Dataset

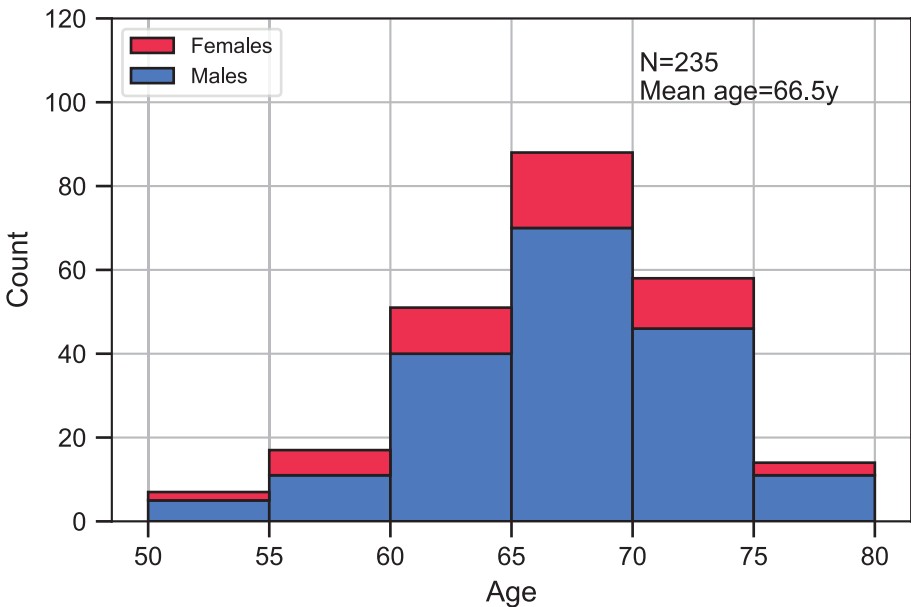

**Appendix 1—figure 3.** Sample size distribution in the analysis that investigated type 2 diabetes mellitus (T2DM) duration as the main factor. The values shown represent one out of the three duration groups (healthy controls [HCs], T2DM+ for 0–9 years, and T2DM+ for 10+ years). Samples across the three duration groups were matched for age, sex, education, and hypertension. Given the three groups, the total sample size was three times what is shown here. The statistical analysis included T2DM+ samples only (for which duration was defined), HCs were included for visual representation only. The bars are stacked.

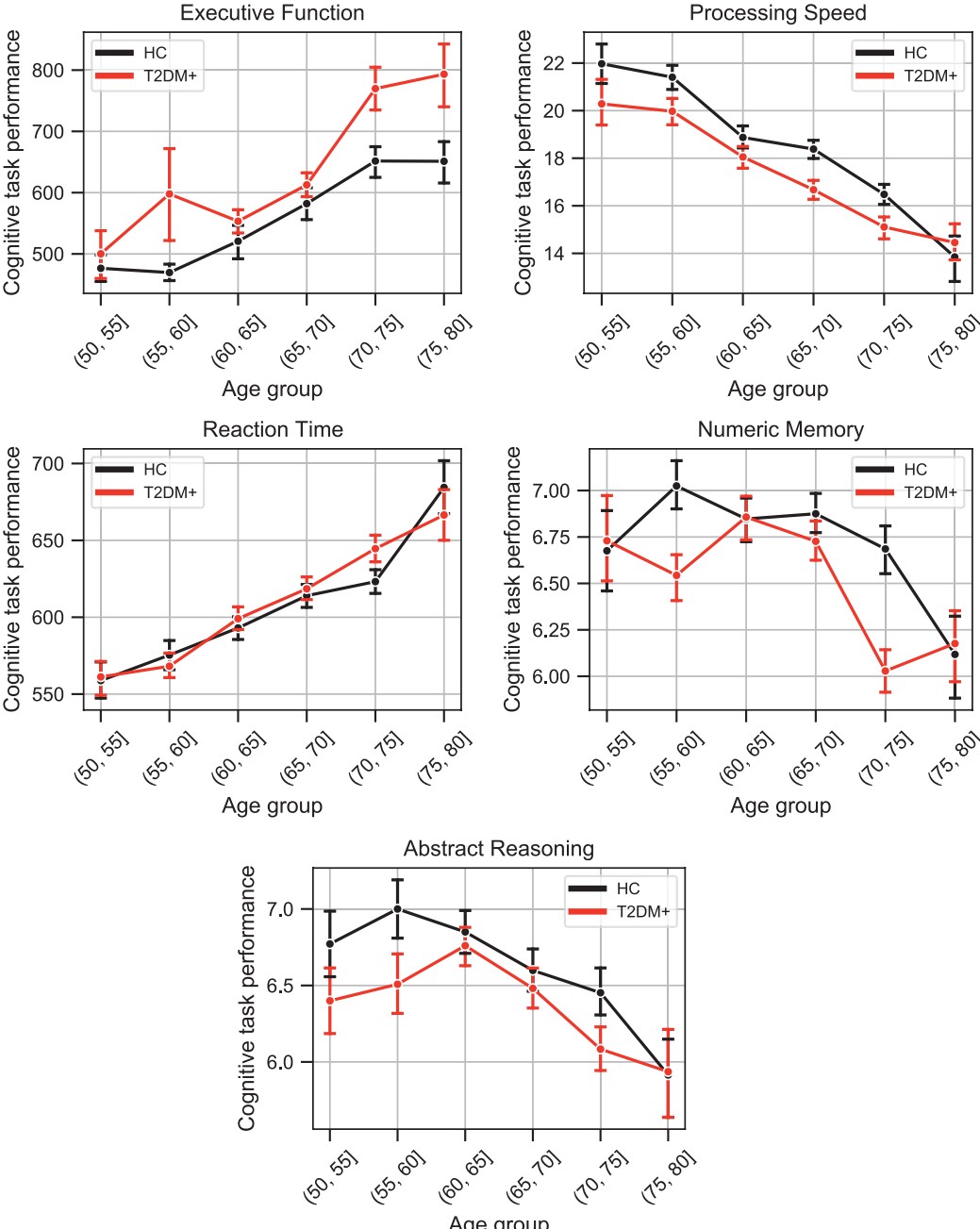

Cognitive Performance Trends across Age and T2DM Status, UK Biobank Dataset

**Appendix 1—figure 4.** Plots representing trends across age and type 2 diabetes mellitus (T2DM) for each of the five investigated cognitive domains. For *executive function* and *reaction time*, lower scores represented better performance. The linear trends across age were robust and justified modeling age as a linear factor. The apparent deviation from linear trends in the youngest and oldest age groups is explained by markedly smaller sample sizes in those age groups (see underlying sample sizes in *Appendix 1—figure 1C*).

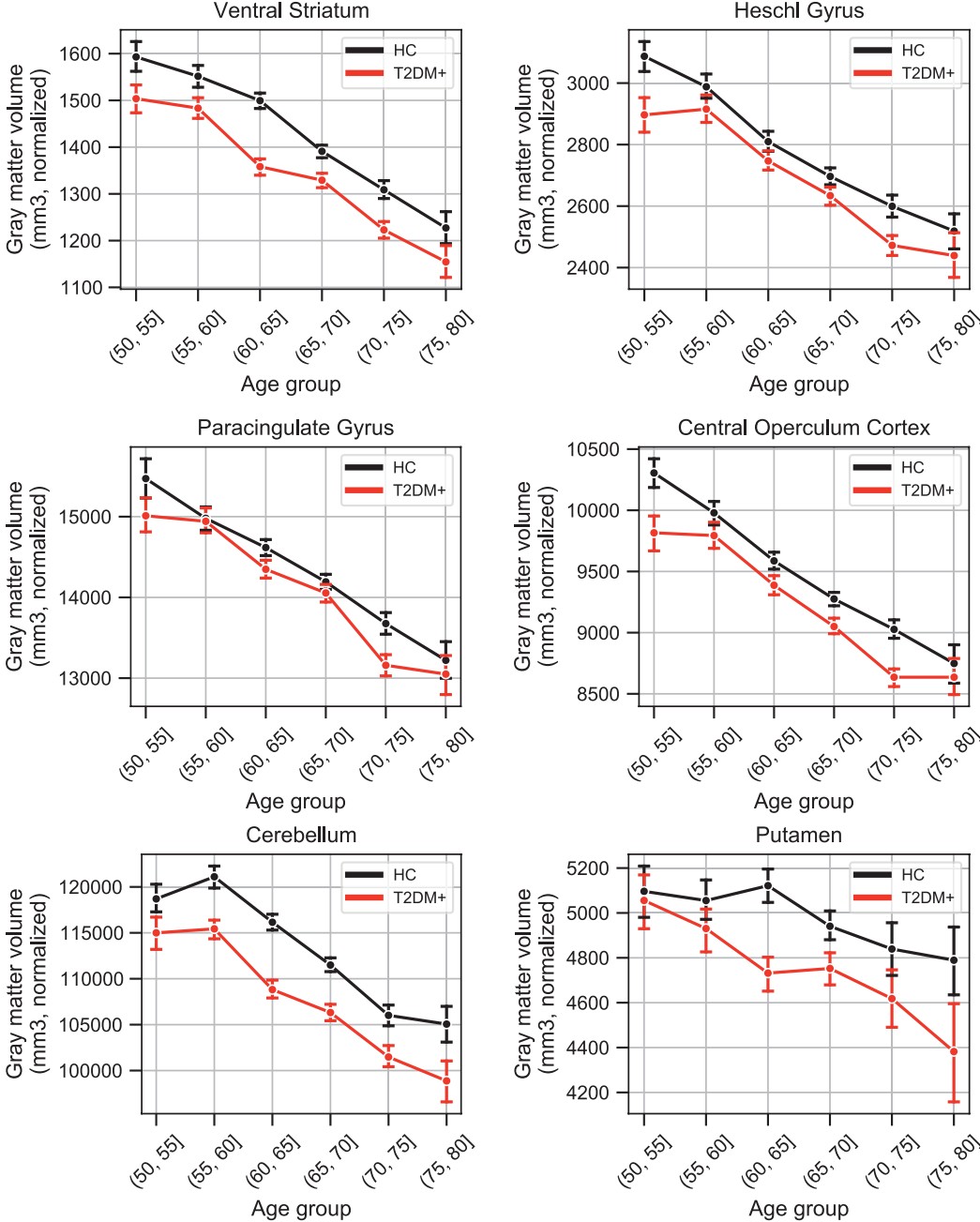

**Appendix 1—figure 5.** Plots representing trends across age and type 2 diabetes mellitus (T2DM) for six regions that exhibited the strongest trends in gray matter volume changes. The linear trends across age were robust and justified modeling age as a linear factor. The apparent deviation from linear trends in the youngest and oldest age groups is explained by markedly smaller sample sizes in those age groups (see underlying sample sizes in *Appendix 1—figure 1A*).

Brain Activation (ALFF) Reorganization Trends across Age and T2DM Status, UK Biobank Dataset

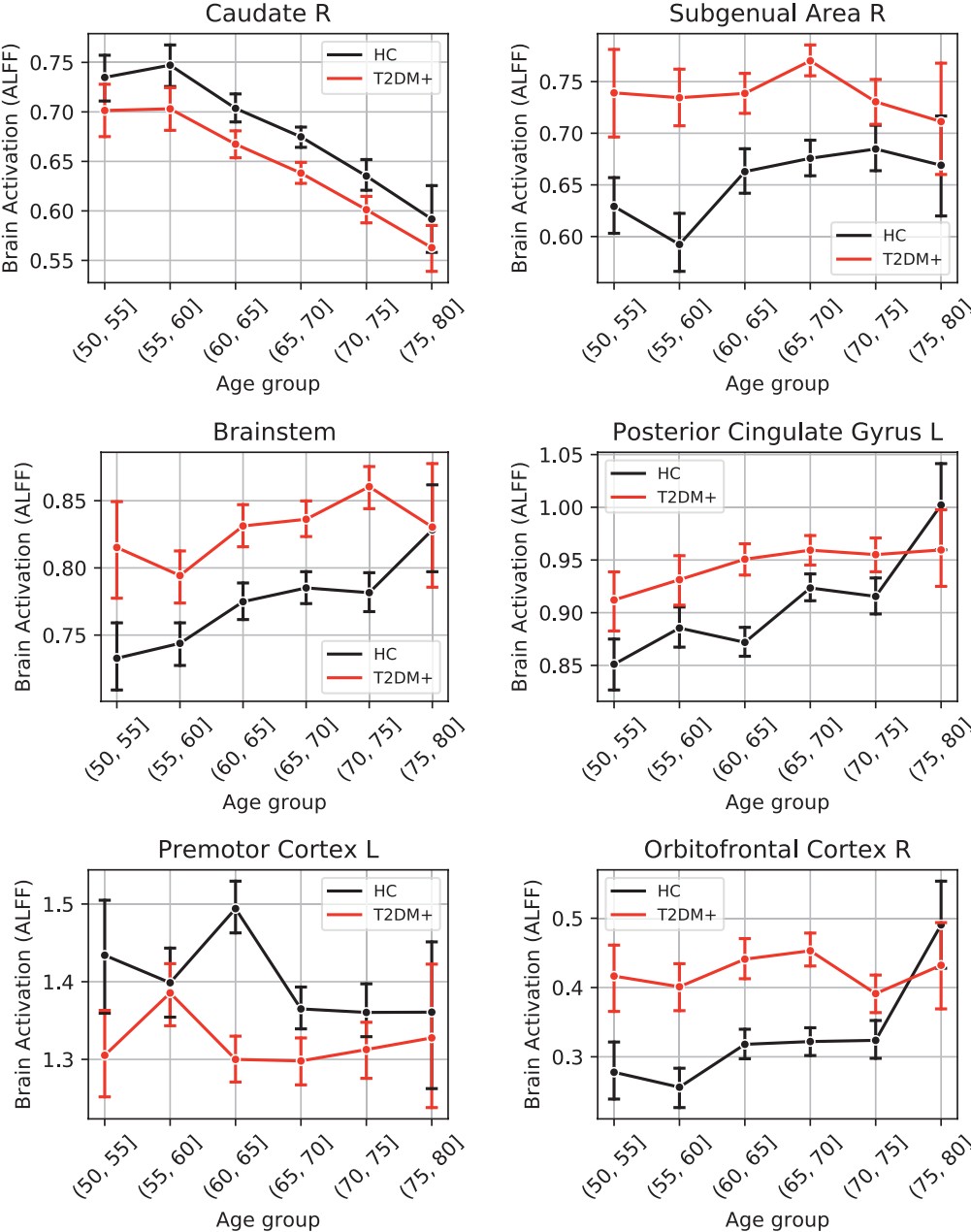

**Appendix 1—figure 6.** Lineplots representing trends across age and type 2 diabetes mellitus (T2DM) for six regions that exhibited the strongest trends in brain activation (amplitude of low-frequency fluctuation [ALFF]). The linear trends across age were robust and justified modeling age as a linear factor. The apparent deviation from linear trends in the youngest and oldest age groups is explained by markedly sample sizes in those age groups (see underlying sample sizes in *Appendix 1—figure 1B*).

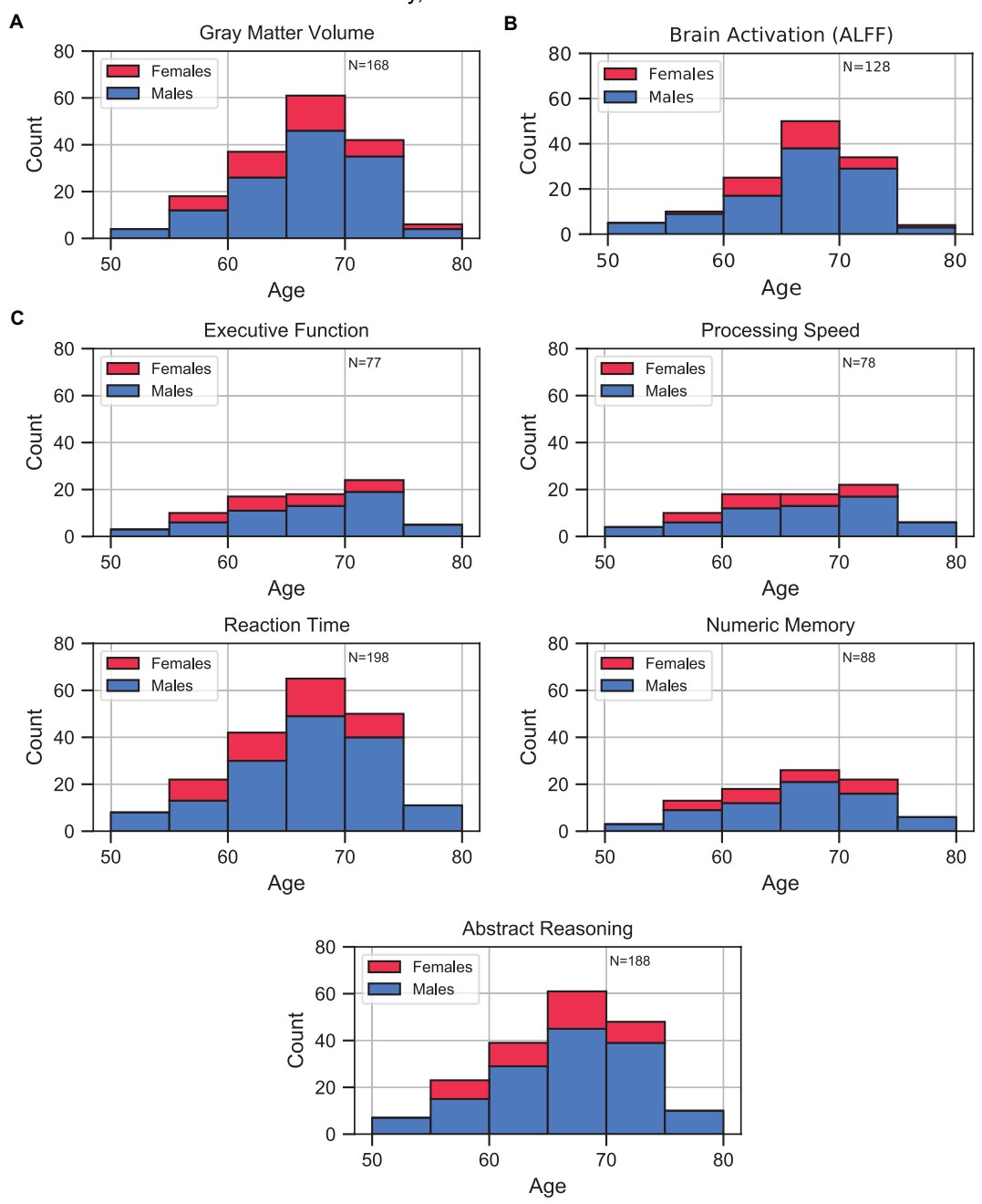

Sample Sizes for Metformin-Related Analyses from UK Biobank,
Medicated Only, to be Matched with Unmedicated

**Appendix 1—figure 7.** Sample size distributions in metformin medication status-related analyses. The values shown represent metformin medicated type 2 diabetes mellitus (T2DM)+ only, these samples were matched with an equivalent number of unmedicated T2DM+ samples, resulting in twice the number of samples in every analysis. Samples were matched for age, sex, education, and hypertension. The bars are stacked. (**A**) Gray matter volume analyses, (**B**) brain activation analyses, and (**C**) cognition analyses (five domains).

