## [Editor Report]

This work emphasizes the role of diabetes in brain aging and cognitive functions that is a critical gap that needs to be filled due to the increasing trend in the prevalence of diabetes around the world. This article provides valuable information about specific brain regions altered during aging and diabetes. Further, this article reports how T2DM accelerates the aging-associated decline in cognition and brain function. Extensive analysis of human datasets and comparison with published data from other researchers support the conclusion of this study; however, certain diabetic interventions that do not rescue brain damage need further validation.

---

## [Decision Letter]

**Decision letter after peer review:**

Thank you for submitting your article "Type 2 diabetes mellitus accelerates brain aging and cognitive decline: complementary findings from UK Biobank and meta-analyses." for consideration by *eLife*. Your article has been reviewed by 2 peer reviewers, and the evaluation has been overseen by a Reviewing Editor and Carlos Isales as the Senior Editor. The reviewers have opted to remain anonymous.

In this timely study the authors explore the impact of age and diabetes status on cognition. The study leverages the UK Biobank data set and validates findings using a meta-analysis of the relevant clinical literature. The major finding is that Type II diabetes is associated with changes in brain structure and cognition similarly to those identified in aging but occurring earlier in individuals with Type II diabetes.

Please respond to all the issues raised by the reviewers.

Essential Revisions (for the authors):

1. Sexual dimporphism.

2. Control group- is it free of diabtes or pre- diabetics as well.

3. Exclusion of T1DM based on age cutoff is problematic as well because people can develop T1DM even after 20 years of age.

4. It takes many years before people develop T2DM, and transition from a normal glucose metabolism, to insulin resistance, prediabetes, diabetes; associated with these changes there are many other associated inflammatory, metabolic and molecular changes of systemic and cerebral blood vessels. Authors should discuss this point in their paper.

5. The duration of diabetes and metformin treatment.

*Reviewer #1 (Recommendations for the authors):*

1. Please provide more information regarding how the mid-life or lat*e-*life onset of T2DM influences cognition and neurodegeneration in the analyzed dataset. The duration of diabetes must be considered along with age, sex, and education while performing paired analysis. Provide the average age of subjects selected in T2DM and HC groups? Also, include the minimum and maximum age of subjects included in each group for the assays.

2. Since sex hormones play an essential role in T2DM and neurodegeneration, I would recommend the authors separate the cognition data based on sex difference and present it as a figure to show how sex differences and their association with T2DM influence cognitive performance.

3. Does early diabetic intervention limit neurodegeneration and cognition decline in the analyzed dataset? What is the disease severity between the metformin-treated and the untreated groups?

4. To compare the meta-analysis with UK Biobank data analysis, the parameters used for analysis should be similar. Is the meta-analysis for cognitive performance include sex-based comparison? There is no information regarding this in the methods section. Please clarify

5. Please check the dataset, for other disease conditions or co-morbidities that could potentially affect the studied cognition domain or brain structure and activity. Because conditions like hypertension that are usual in aged and diabetic conditions can influence the observed data.

*Reviewer #2 (Recommendations for the authors):*

Strengths of the study include

a) the use of the UK Biobank,

b) identification of overlap between the age-sensitive brain structural and functional changes with those in the T2DM group,

c) demonstration that the duration of T2DM impacts the extent of change,

d) interesting outcome of the ALFF pointing to reorganization rather than simple decline in "activity" by brain region,

e) the validation using the meta-analysis and neuroquery combination,

f) the regression analysis showing congruence for structural outcomes and for cognitive outcomes between the UK Biobank and Meta-analysis approaches.

Moderate concerns include

a) The primary focus here is on T2DM, for which (according to the methods) the comparisons to healthy controls were pairwise matched by age, sex, education. The actual number of subjects used is unclear: The UK Biobank dataset included cognitive and neuroimaging data (N=26,125) including 1,270 T2DM and 24,855 HC. T2DM cohort is 5% the size of the HC. What was the actual reference analysis? Were 1270 matched HC used as the reference or were all HC included – how did this impact variance in the measures?

b) Given the cross-sectional nature of the work differences in composition of the cohort at the various times points and the numbers contributing to each of the increments would be helpful. Were the numbers of data points equivalent across the 50-75 years of age?

c) Can the authors provide an explanation as to why males and females were combined?

d) It would be helpful to understand how other potentially important variables such as BMI contributed in the analysis.

e) The declines in the various parameters with age are quite modest, making those identified in the T2DM all the more striking. In the narrative declines per year are described but actual age is not mentioned and does not appear at all until later in the manuscript (Figure 4) and (Figure S1). The data are shown as 5-year increments from 50-75 years. It is unclear what happened before this time point – is there a period during adulthood when volume remains steady? Appreciating the ~8.3 to ~7.6 GMV voxel count over the 25 year time frame – it would be helpful to understand if the declines are detected from 50 years or earlier.

f) Was sex or BMI a contributing factor in either the T2DM or the HC outcomes?

g) The final conclusion of the abstract "As such, neuroimaging-based biomarkers may provide a valuable adjunctive measure of T2DM progression and treatment efficacy based on neurological effects" is not supported. Actual comparisons to established clinical assessments of glucoregulatory impairment were not conducted (effect size and sensitivity), and apart from the obvious considerations of cost and accessibility, this comment seems unnecessary and detracts from the work.

Comments:

a) The authors point to "accelerated" cognitive aging but it seems the slopes of decline are rather similar for aging and for T2DM – what differs is the starting value at age 50. It is semantics I suppose but accelerated seems a better fit if the pace of aging is increased, i.e. a different slope. It is appreciated that premature onset of aging phenotypes has in the field been described as accelerated but often it is also accompanied by a quicker rate of decline. What seems to be happening here is premature age-related decline.

b) Speculation about "switch to less metabolically expensive networks" is unwarranted. In the absence of glucose uptake measures "diminished access to glucose" is not tested. Even with those data, inferences about the metabolism of glucose cannot be made as PET tells nothing about the fate of glucose. This argument ignores the possibility of a role of insulin, lipid signaling, or even inflammatory processes that might equally be speculated upon.

c) It is somewhat alarming that while 559 of the T2MD were treated with metformin for their condition, 473 were unmedicated. Is this usual?

---

## [Author Response]

Essential revisions:1. Sexual dimporphism.

In our original submission, we had already excluded the possibility that our results were solely due to sexual dimorphism, as we matched for sex in all analyses in the UK Biobank (every female with T2DM was matched with a female HC control, and equivalently for males). When investigating age, effects were derived from a combination of equally distributed *female only* and *male only* subsamples. However, we agree with the reviewers that differences due to sexual dimorphism in the UK Biobank are relevant and should be reported. Edits can be found on lines 192-195, 358-375, and 385-390.

To eliminate interaction effects stemming from the menopausal transition process in females, we excluded all female subjects who did not report menopause or reported hormone replacement therapy. As the distribution of the raw sample was strongly skewed towards ages >60 years, these exclusions did not result in significant loss of sample size.

To address sexual dimorphism, we compared females and males across all three investigated sets of features: cognitive performance, atrophy (gray matter volume), and brain activity (ALFF). For all three, we found significant differences between males and females, which are now reported in Figure 1—figure supplement 1, Figure 2—figure supplement 3 and Figure 3—figure supplement 3.

In addition, we investigated the interaction between sex and our main factors of interest: T2DM and age, by performing all main analyses independently for females only and for males only. We found that, although age-related effects were equivalent in females and males, T2DM effects were more pronounced in males, even if matched for disease duration (see Figure 1—figure supplement 2, Figure 2—figure supplement 4, Figure 2—figure supplement 5, Figure 3—figure supplement 4). This manifested not only in reduced statistical power in females due to smaller sample sizes, but also in smaller mean differences between HC and T2DM in females. We discuss these results and a potential explanation in the discussion on lines 385-390. Our second level analyses correlating effects associated with T2DM and age also showed stronger results in the *male only* subsample (Figure 2—figure supplements 6, 7 and Figure 3—figure supplements 5, 6). However, given that both males and females show the same effects (the differences are of degree, but not of kind), we also present our results from the combined sample.

2. Control group- is it free of diabtes or pre- diabetics as well.

The groups were defined as those that had vs. had not been diagnosed with diabetes by a doctor (“Diabetes diagnosed by doctor”, UK Biobank datafield #2443). While not perfect, this was the best available such feature in the dataset. We acknowledge this limitation in the discussion on lines 430-435. However, it should reassure the reviewer to note that, if there were prediabetics in the control group, this would have weakened the effect size for the comparison, rather than strengthened it. Therefore, the fact that the control group may have contained prediabetics, and yet we still found marked statistical differences between that group and those diagnosed with diabetes, suggests that the results are very robust and might even be more robust than reported.

3. Exclusion of T1DM based on age cutoff is problematic as well because people can develop T1DM even after 20 years of age.

According to a recent study using the same UK Biobank dataset, investigating the distribution of age at diagnosis of T1DM based on genetic markers, only ~1.8% of diagnoses above the age of 40 involved T1DM [1]. Thus, we re-analyzed the data, choosing 40 years as the new cutoff in terms of age at diagnosis.

Reviewer #1 (Recommendations for the authors):1. Please provide more information regarding how the mid-life or lateLife onset of T2DM influences cognition and neurodegeneration in the analyzed dataset. The duration of diabetes must be considered along with age, sex, and education while performing paired analysis. Provide the average age of subjects selected in T2DM and HC groups? Also, include the minimum and maximum age of subjects included in each group for the assays.

We have made edits to the manuscript (line 106) and have added several new figures to clarify. Exact sample sizes and corresponding distributions across age are shown on Appendix figures 1, 2, 3, and 7 for all analyses. Regarding the actual age distributions, all samples were aged 50-80 years, the average age typically being ~65 years across analyses. Given this distribution and the T1 diabetes-related age of onset cutoff at 40 years, our analyses were specific to cases with mid-life and lat*e*-life onset of T2DM. Our analysis of disease duration with respect to gray matter volume did not reveal different outcomes for cases of mid-life vs lat*e*-life onset of T2DM, while matching for duration (see Figure 4). The significant association between disease duration and neurodegeneration appears to be consistent in its extent across the 50-80y age range.

2. Since sex hormones play an essential role in T2DM and neurodegeneration, I would recommend the authors separate the cognition data based on sex difference and present it as a figure to show how sex differences and their association with T2DM influence cognitive performance.

As described above (Essential Comment #1), we re-analyzed all UK Biobank data to investigate the interaction between sexual dimorphism and T2DM. We now present these results in the manuscript (lines 358-375) and show relevant results on Figure 1—figure supplement 2.

3. Does early diabetic intervention limit neurodegeneration and cognition decline in the analyzed dataset? What is the disease severity between the metformin-treated and the untreated groups?

Unfortunately, the dataset did not include sufficient information to evaluate this question. Accordingly, we indicate in the Discussion section (lines 430-435), that we did not have access to appropriate biomarkers such as blood-based measures to assess disease severity between metformin treated and untreated groups. We did control for BMI as the best proxy available, but we did not observe marked differences in this measure: the average BMI was 29.1 and 30.6 for unmedicated and medicated samples, respectively.

4. To compare the meta-analysis with UK Biobank data analysis, the parameters used for analysis should be similar. Is the meta-analysis for cognitive performance include sex-based comparison? There is no information regarding this in the methods section. Please clarify

Our meta-analysis of cognitive performance was unable to perform a sex-based comparison, as the underlying papers that our data were derived from did not perform sex-matching across their subject pools and did not provide access to the individual subject-level data that would be required to control for sex in our analyses.

5. Please check the dataset, for other disease conditions or co-morbidities that could potentially affect the studied cognition domain or brain structure and activity. Because conditions like hypertension that are usual in aged and diabetic conditions can influence the observed data.

To address this issue, we now have quantified hypertension based on peripheral blood pressure measurements for all samples, as described in the Methods section (lines 121-123). Consistent with literature, we observed hypertension to influence gray matter volume in several brain regions [2] as well as cognition (short-term memory and abstract reasoning). Since hypertension moderately correlated with T2DM+ status, we repeated all analyses in the UK Biobank by matching for hypertension (along with age, sex and education), to mitigate the variable as a confound.

Reviewer #2 (Recommendations for the authors):a) The primary focus here is on T2DM, for which (according to the methods) the comparisons to healthy controls were pairwise matched by age, sex, education. The actual number of subjects used is unclear: The UK Biobank dataset included cognitive and neuroimaging data (N=26,125) including 1,270 T2DM and 24,855 HC. T2DM cohort is 5% the size of the HC. What was the actual reference analysis? Were 1270 matched HC used as the reference or were all HC included – how did this impact variance in the measures?

This information is now clarified in the manuscript, primarily in Appendix figures 1, 2, and 3. In our T2DM vs HC comparisons, we used all eligible T2DM+ samples and matched each one of them with a HC from the total sample, in a 1:1 ratio. For instance, in the case of gray matter volume, we compared 821 T2DM+ to 821 HC which were matched for age, sex, education and hypertension, but otherwise randomly picked from the total sample. Consequently, this resulted in numerous HC samples being discarded. When addressing effects associated with age, we investigated within HC only, resulting in significantly larger sample sizes and statistical power.

b) Given the cross-sectional nature of the work differences in composition of the cohort at the various times points and the numbers contributing to each of the increments would be helpful. Were the numbers of data points equivalent across the 50-75 years of age?

Please see Appendix figures 1, 2, 3, and 7, which now clearly indicate the distribution of samples across the age span, separately for every analysis we performed in the UK Biobank.

c) Can the authors provide an explanation as to why males and females were combined?

We have now re-analyzed all UK Biobank data to investigate the interaction between sex and T2DM status. We refer the reviewer to our response given to an earlier comment (Essential Comment #1) where we explain our findings in detail. Please note that all our UK Biobank analyses matched for sex, to exclude it as a confounder. With respect to sex specific T2DM related effects, we did not detect qualitatively different results when performing analyses in *females only* vs in *males only*, which justified our decision to present the main results combining sexes. Sex-specific results are now presented in Supplemental Figures.

d) It would be helpful to understand how other potentially important variables such as BMI contributed in the analysis.

In response to this comment, we re-performed all our analyses with BMI as an additional covariate and found BMI to be a significant predictor for gray matter volume, but not for cognition or brain function. As for structure, we were unable to dissociate BMI and T2DM associated effects given the strong collinearity between increased BMI and T2DM+ status. The collinearity was so strong that even matching was deemed impractical due to significant sample size loss (lack of T2DM+ with low BMI).

e) The declines in the various parameters with age are quite modest, making those identified in the T2DM all the more striking. In the narrative declines per year are described but actual age is not mentioned and does not appear at all until later in the manuscript (Figure 4) and (Figure S1). The data are shown as 5-year increments from 50-75 years. It is unclear what happened before this time point – is there a period during adulthood when volume remains steady? Appreciating the ~8.3 to ~7.6 GMV voxel count over the 25 year time frame – it would be helpful to understand if the declines are detected from 50 years or earlier.

The ~0.5-1% decline per year we identified with respect to age (as shown in Figure 1 for cognition and Figure 2 for gray matter volume) were indeed specific to 50 years and above. Less than 1% of our sample was younger than 48 years and none was younger than 44 years. This precludes the evaluation of aging-related neurocognitive deterioration prior to age 50.

f) Was sex or BMI a contributing factor in either the T2DM or the HC outcomes?

Sex was a statistically significant contributing factor across all our features; consequently, we matched for it. Please once again refer to our earlier response to Essential Comment #1. BMI, as described in our response above to another comment above, was also significant but only in the context of gray matter volume.

g) The final conclusion of the abstract "As such, neuroimaging-based biomarkers may provide a valuable adjunctive measure of T2DM progression and treatment efficacy based on neurological effects" is not supported. Actual comparisons to established clinical assessments of glucoregulatory impairment were not conducted (effect size and sensitivity), and apart from the obvious considerations of cost and accessibility, this comment seems unnecessary and detracts from the work.

We respect the reviewer’s perspective. However, our remark occurs at the end of the Discussion section in the context of a direction for future clinical research. We believe that this research is warranted based on the widespread neurodegenerative effects that we identified in patients with T2DM.

Comments:a) The authors point to "accelerated" cognitive aging but it seems the slopes of decline are rather similar for aging and for T2DM – what differs is the starting value at age 50. It is semantics I suppose but accelerated seems a better fit if the pace of aging is increased, i.e. a different slope. It is appreciated that premature onset of aging phenotypes has in the field been described as accelerated but often it is also accompanied by a quicker rate of decline. What seems to be happening here is premature age-related decline.

This is more clearly represented in the analyses where we break up the T2DM+ cohort with respect to disease chronicity. Specifically, in Figure 4 we show that as disease duration increases, the deviation between T2DM+ and HC increases (acceleration of 0.26% ± 0.14%).

b) Speculation about "switch to less metabolically expensive networks" is unwarranted. In the absence of glucose uptake measures "diminished access to glucose" is not tested. Even with those data, inferences about the metabolism of glucose cannot be made as PET tells nothing about the fate of glucose. This argument ignores the possibility of a role of insulin, lipid signaling, or even inflammatory processes that might equally be speculated upon.

This claim is based on existing literature [3, 4], providing evidence not only for inhomogeneity among brain regions in terms of metabolic requirements and efficiency. Subcortical regions and the cerebellum were found to be metabolically cheaper, which is relevant to the brain activity patterns we observed in relation to T2DM and age: brain activity manifested a general shift from cortical to subcortical regions (Figure 3) [3].

c) It is somewhat alarming that while 559 of the T2MD were treated with metformin for their condition, 473 were unmedicated. Is this usual?

Half of unmedicated subjects were diagnosed less than five years earlier (25% within two years), whereas for medicated subjects, average duration was twice as high. This suggests that a significant number of subjects might not have been diagnosed for sufficient time to start medication. In addition, our sample of 498 subjects who reported taking metformin included those only who took metformin exclusively (we excluded those taking multiple medications to treat their diabetes). There were an additional ~150 subjects who were excluded due to polypharmacy, further raising the ratio.

References:

[1]: Thomas, Nicholas J., et al. "Frequency and phenotype of type 1 diabetes in the first six decades of life: a cross-sectional, genetically stratified survival analysis from UK Biobank." *The Lancet Diabetes and Endocrinology* 6.2 (2018): 122-129.

[2]: Gianaros, Peter J., et al. "Higher blood pressure predicts lower regional grey matter volume: Consequences on short-term information processing." *Neuroimage* 31.2 (2006): 754-765.

[3]: Tomasi, Dardo, Gene-Jack Wang, and Nora D. Volkow. "Energetic cost of brain functional connectivity." *Proceedings of the National Academy of Sciences* 110.33 (2013): 13642-13647.

[4]: Shokri-Kojori, Ehsan, et al. "Correspondence between cerebral glucose metabolism and BOLD reveals relative power and cost in human brain." *Nature communications* 10.1 (2019): 1-12.